To all reviewers:

Thank you for your comments and suggestions to improve this work. We feel that in using the concept as defined in our paper, models can be simplified, with more transparent and better understood outcomes. Additionally, with the findings of our study, models can be made more efficient as to training time. We expand on this below in italics.

Reviewer ZHzr:

1. The paper defines "node stability" as

    > "the number of epochs wherein nodes held their rank in terms of weight value compared to their rank on the last epoch".

By this definition, stability will always reach 100% in the last epoch, and little can be determined from the "exponential-looking" curve of DNNs stability. The high increase in the end is, by definition, inevitable, and does not really indicate the network's weights are "more stable". However, the underlying idea of measuring NN stability is interesting. Perhaps it would be better measured as stability of rank between each two consecutive epochs, and not using the last epoch as an anchor.

*Yes, the stability must always reach 100 percent by our definition. That, however, is not critical. The issue is when each node stabilizes. That process, over time, forms a tree structure. The importance of the definition is in establishing decision tree behavior and the implications thereof in model simplification.*

2. Equivalence between node significance and its weight value is unfounded (although proving this would be interesting).

Throughout the paper the absolute weight value of a node is used equivalently to its "significance" or "influence in performance". This is actually not backed up by theory or by empirical results. A node with high weight value may consistently receive inputs with very low absolute value (depending on the dataset), and a node with low weight value may receive inputs with high absolute value. The relationship between absolute weight value and the influence of that node on model performance is never tested in the paper.

It would indeed be interesting to have an experiment where each node (or group of nodes) is pruned at a time, and performance of the pruned model is assessed, so we could draw some empirical conclusions on whether intermediate nodes are indeed less valuable than higher rank nodes.

*That is intrinsic in the definition of node rank. If the rank is maintained, the weights must be of high absolute value. If the weight absolute value is high for a node, that node must have been fed high valued weights, or it would lose its rank along training. And a high weight value must be more influential in model outcomes.*

3. The use of the term "decision tree", both in the title and throughout the paper, is a bit misleading. No DTs are actually used anywhere in the paper, it is continuously used as (somewhat arbitrary) analogy. Nodes with high absolute weight value do not behave as root nodes do in DTs (or at least this is not proven in the paper).

*This needs a much better explanation in the text, as we are not using a decision tree. We are remarking that the neuron stabilization forms the shape of a tree and therefore forms the decision tree structure. Below is one possible correction to the misunderstanding.*

*To strengthen the decision tree behavior claim above, presenting an illustration from a fictitious case may be of use. In the illustration, a node configuration as to rank stability of the flattened focus layer after ordering according to node weight may be expressed as in Figure X. Figure X shows two root nodes stabilized in the first of four epochs and exactly two additional child nodes per parent node stabilized after each subsequent epoch, showing a shape resembling a decision tree (Step 1). The tree behavior observed in this study deviated from the illustration below but held the same general pattern from this fictitious example. Additionally, Step 2 in Figure X shows a different perspective of the decision tree layout, where the extremes of the ordered flattened focus layer are stable nodes of either highly negative or highly positive values, which also have the highest rank stabilities. In Step 3, a bar chart is presented as a visualization aid for Step 2. It is worth mentioning that in the fictitious example, the root nodes stabilize at the first epoch. This is unlikely to be the case in the actual observations of this study. It is, however, the case in the illustration due to the limited number of epochs used in the interest of saving space. Additionally, no groupings were necessary in the example, as only 11 nodes are illustrated.*

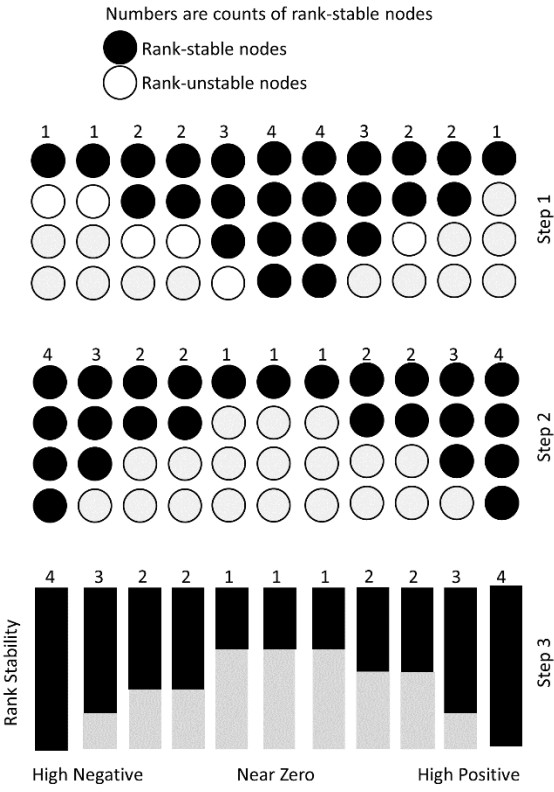

*Figure X. Tree behavior according to node rank stability during a fictitious neural network training progression; number above bars are rank stability values.*

4. The reader is left wandering what is a possible outcome of discovering "decision tree behavior" in NNs? How can this be used? A lot of hedging is used throughout the paper with respect to possible utility of its results ("may", "might", "can", "possibly", etc.)

*Perhaps the main contribution of this study's findings is in model simplification. With a simplified model, outcomes thereof may be made more transparent and understanding. An additional contribution is in training efficacy, as the tree behavior may aid in determining early stopping points during training.*

Reviewer b5kq:

Major issues:

- The actual work presented in the paper is quite different from the title, the introduction, related work, and most of the abstract. While the narrative on XAI is interesting, the only connection with the work is perhaps in future work. This may perhaps be due to a misuse of the term XAI / Explainable Artificial Intelligence, but even so, the paper does not provide any way to "reap the fruits of decision trees" either.

*Yes, and we see that being the case. This, however, can be addressed by connecting XAI to model simplification more clearly, the main contribution of this study's findings. With a simplified model, outcomes thereof may be made more transparent and understanding. An additional contribution is in training efficacy, as the tree behavior may aid in determining early stopping points during training*

- In particular, what do the results tell us about how a neural network has made its decisions?

*The results show resemblance to a decision tree, with its implications in model simplification. We are highlighting that the neuron stabilization is in the shape of a tree and therefore forms the decision tree structure.*

- The stability criterion set to 90% accuracy seems fairly arbitrary, and it seems it should rather be task specific (what if a network cannot achieve 90% accuracy on a task?). In general, the notion of "stable models" in the paper is confusing; I think the authors tried to determine when the network stopped learning (so that weights ranks would be finalized), but in that case it would have been better to calculate the magnitude of changes to the total weights vector directly, and/or use learning rate annealing to guarantee convergence (and thus rank stability).

*Yes, arbitrary. However, it is hard to envision a successful neural network result being adopted with less than 90% accuracy. The value was selected as a minimum criterion. That figure was chosen, again, to highlight decision tree behavior. If the model cannot achieve acceptable accuracy, it may not be possible to simplify it using what we proposed.*

- The authors stress the importance of analyzing individual units in a neural network (in the abstract and introduction), yet in the actual work performed they ignore all units except those of the last hidden layer of the network.

*The individual units are the aim. The groupings are to enable better visualization. We can clarify that.*

- I am not sure how weights are ranked when multiple output units are used instead of a single one: are all connections from all units in the last hidden layer and ALL the output units considered? Or is the analysis performed separately for each output unit?

*It can be clarified that it is the output weight to the softmax that is the focus.*

- The paper is mostly related to the purported 'decision tree dynamics of neural networks', yet the authors never really explain what this means. It is perhaps a concept known in literature that I am not aware of (if so, it should be cited in the paper), but it would be useful to explain it clearly. The concept is developed a bit further at the end of the paper, but still not clearly.

*Yes, pasted below is what we answered above as a possible explanation. We are stating that neuron stabilization is shaped like a tree, forming the decision tree structure*

*To strengthen the decision tree behavior claim above, presenting an illustration from a fictitious case may be of use. In the illustration, a node configuration as to rank stability of the flattened focus layer after ordering according to node weight may be expressed as in Figure X. Figure X shows two root nodes stabilized in the first of four epochs and exactly two additional child nodes per parent node stabilized after each subsequent epoch, showing a shape resembling a decision tree (Step 1). The tree behavior observed in this study deviated from the illustration below but held the same general pattern from this fictitious example. Additionally, Step 2 in Figure X shows a different perspective of the decision tree layout, where the extremes of the ordered flattened focus layer are stable nodes of either highly negative or highly positive values, which also have the highest rank stabilities. In Step 3, a bar chart is presented as a visualization aid for Step 2. It is worth mentioning that in the fictitious example, the root nodes stabilize at the first epoch. This is unlikely to be the case in the actual observations of this study. It is, however, the case in the illustration due to the limited number of epochs used in the interest of saving space. Additionally, no groupings were necessary in the example, as only 11 nodes are illustrated.*

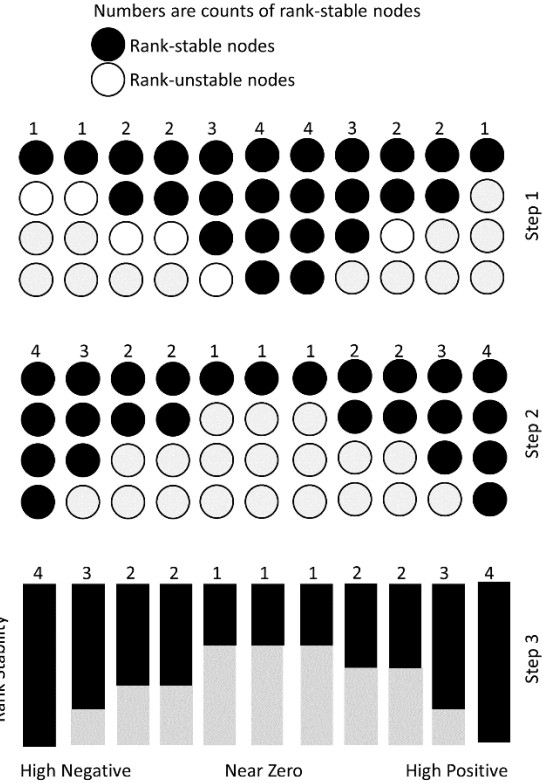

*Figure X. Tree behavior according to node rank stability during a fictitious neural network training progression; number above bars are rank stability values.*

- I also don't understand how node rank stability would be related to the decision tree behavior of the network? Is it because some of the nodes that get 'fixed' early are made to correspond to early decision nodes in a decision tree, etc?

*Yes, this can again be made clear as above. That process, over time, forms a tree structure. The importance of the definition is in establishing decision tree behavior and the implications thereof in model simplification.*

- I am not entirely convinced that the results demonstrate 'decision tree behavior' in neural networks; it seems to me that the observed dynamics are more likely due to gradient descent in high dimensional spaces. The authors write that "Such nodes [groups] are analogous to the concept of decision tree roots. The timing when those weights and their ranks were established might have varied among individual runs but did show a clear pattern according to groups." So, the units in those groups may learn earlier or later than others, but at runtime they are all run at the same time, non-hierchically. I understand it may be used as a type of combinatorial code perhaps, but that is not really assessed (e.g.,

can we actually see what the neurons in the earlier vs later groups encode?). I would be grateful again if the authors could explain the concept more clearly.

*We agree. What needs to be made clear is the objective, ie, model simplification. Gradient descent behavior would be more likely dependent on the input data, i. e., tweaking the input and noticing how the change is represented in the neuron space. Our study does not depend on the input data to reveal the findings. The decision tree behavior, as we show, is always represented in the neuron space, independently of manipulation in the input data.*

- In any case, *I suspect you would see the same exact behavior even in simple logistic regression* (for example, inputs -in this case corresponding to the last hidden layer- more correlated to the classification task would increase in weight earlier, leading to a larger MRS; isn't this what happens in the paper? Wouldn't it be easier to study it in this or similar simpler case first?). This experiment would be useful even if the dynamics observed were found to be different, as it could be used to make a stronger argument.

*This again became an issue because of our unclear details on the objectives. The contention we are making is that node importance in determining model outcomes during training follows a tree behavior. Early nodes (root and parents) are also the most important in determining model outcomes.*

Minor issues:

- Clarity of the text can be improved greatly (mostly organization of the text, but also some language issues and misuse of technical terms); for example "especially when realizing that weight values may not be as random as previously thought", or "Node analysis consisted of an assessment of whether the progression of nodes from the initial run to stabilization could be modeled by a decision tree, potentially identifying the most significant nodes in determining model accuracy early in the training process.".

*We have revisited grammatical/unclear sections of the manuscript.*

- The distinction between 'Convolutional' vs 'Deep Neural Networks' is not entirely correct. While one may imagine a very shallow CNN, they are in practice always deep. In particular, the CNN used in this paper is deeper than the 'DNN' network (3 layers + the same MLP as the other network). Further, what the authors call a DNN is actually a fully-connected MLP.

*This section has been rewritten in light of the above.*

- Evaluation is performed only on 3 very simple datasets, and only 2 network architectures. It would have been more useful to also include realistic architectures (for example, ResNets), and some simple and well understandable ones (e.g., a simple logistic regression / single-layer nn, as I discussed above).

*We did not focus on the input data. We aimed at the internals. Three very distinct input datasets were selected to show that the approach is not dependent on the data feeding the model.*

- In the MLP (called 'DNN' in the paper), the authors mention that each layer has "30x30 nodes"; however, it does not make sense to report the layer sizes like this, as the layers are fully connected (i.e., all spatial arrangement is lost, so whether they are 900 units arranged on a line, a square, or a thorus, it makes no difference).

*Correct. This was done for visualization and ease of understanding.*

- "All experimental runs converged after 150 epochs" -> I think you mean "converged within 150 epochs"?

*Correct, within. We modified the text to reflect this.*

Reviewer cBc4:

Weakness:

1.  The contributions of the paper are not clearly enlisted or self-revealing

    *We feel that the main contributions of this work are primarily in model simplification and second in model training efficiency. With a simplified model, outcomes may be made more transparent and understanding. As for training efficacy, as the tree behavior may aid in determining early stopping points during training.*

2.  The gaps wrt prior art in Related work section should be highlighted instead of mere referring them

    *We rewrote and modified to add gaps in research and how our contributions will assist in filling them.*

3.  Reduction of dataset size to 100 images and then experiments is going south of experimental result validity

    *We need to clarify that the focus is less on the input data as it is on showing decision tree behavior and its implications. We randomly took 100 NMIST images for each class, totaling 1,000 images representative of the 'population' of NMIST images.*

4.  converge to at least 0.90 - why this threshold? any ablation?

    *We will clarify that the threshold is not critical, as long as tree behavior emerges. As for the threshold, it is hard to envision a successful neural network result being adopted with less than 90% accuracy. The value was selected as a minimum criterion. That figure was chosen, again, to highlight decision tree behavior. If the model cannot achieve acceptable accuracy, it may not be possible to simplify it using what we proposed.*

5.  Not clear about the data distribution and balance affecting results

    *We selected three very distinct datasets as input to the models. These data were to ensure variability in inputs and, thus, minimize any possible contention that the tree behavior is an artifact of input data.*

6.  how will it behave under adversarial attacks?

    *The focus, which needs clarification, is model simplification.*

Suggestions:

1.  Abstract should be to the point
2.  The related work section should follow the current trend - topic-wise listing of prior art

3. Dataset section can be reduced and key contributions be elaborated more
4. Clarity needed - what specific problem you are trying to solve and give the logic behind the approach

*We agree and addressed those limitations in the text.*

Miscellaneous:

1. Break long sentences in smaller forms

*We rephrased large sections in the manuscript based on above suggestions.*

2. mimicking an empty, black background -> what about white (1)

*Input data is not the focus*

3. Analyses in this work were based, according to weight magnitude, etc. - many grammar / sentence construction mistakes

*We have made grammatical adjustments.*

4. quasi-Poisson link function - link?

*It is the distribution that applies to count data where the residual variance is larger than the conditional mean, i. e., the quasi-Poisson GLM fits an extra parameter for the over dispersion.*

5. The Tukey's Honest Significance post-hoc GLM - link?

*No, post-hoc analysis*

6. ANOVA - is correct abbrev in place of AOV

*Changed to ANOVA throughout.*

Reviewer 5BfE:

The paper has a clearly defined experiment setup and meaningful spread of experiments to study a method. I think the paper does not sufficiently discuss how to draw conclusions from the proposed visualizations and summarizations. Adding that would help substantiate the authors claim that this way of looking at neural network training does allow for future improvements.

*We agree. Clarifications as to the objectives and conclusions were incorporated.*

The paper proposes a visualization and demonstrates it in various examples, but does not show how to draw practical conclusions from those visualizations to motivate using those visualizations while designing new applications for neural networks.

*With the clarifications above, we can show how decision tree behavior can assist in model simplification and how that will contribute to XAI. Additionally, below is how the visualization we proposed has been better explained in the manuscript, giving a better idea of how a decision tree is inferred from this study's findings.*

*To strengthen the decision tree behavior claim above, presenting an illustration from a fictitious case may be of use. In the illustration, a node configuration as to rank stability of the flattened focus layer after ordering according to node weight may be expressed as in Figure X. Figure X shows two root nodes stabilized in the first of four epochs and exactly two additional child nodes per parent node stabilized after each subsequent epoch, showing a shape resembling a decision tree (Step 1). The tree behavior observed in this study deviated from the illustration below but held the same general pattern from this fictitious example. Additionally, Step 2 in Figure X shows a different perspective of the decision tree layout, where the extremes of the ordered flattened focus layer are stable nodes of either highly negative or highly positive values, which also have the highest rank stabilities. In Step 3, a bar chart is presented as a visualization aid for Step 2. It is worth mentioning that in the fictitious example, the root nodes stabilize at the first epoch. This is unlikely to be the case in the actual observations of this study. It is, however, the case in the illustration due to the limited number of epochs used in the interest of saving space. Additionally, no groupings were necessary in the example, as only 11 nodes are illustrated.*

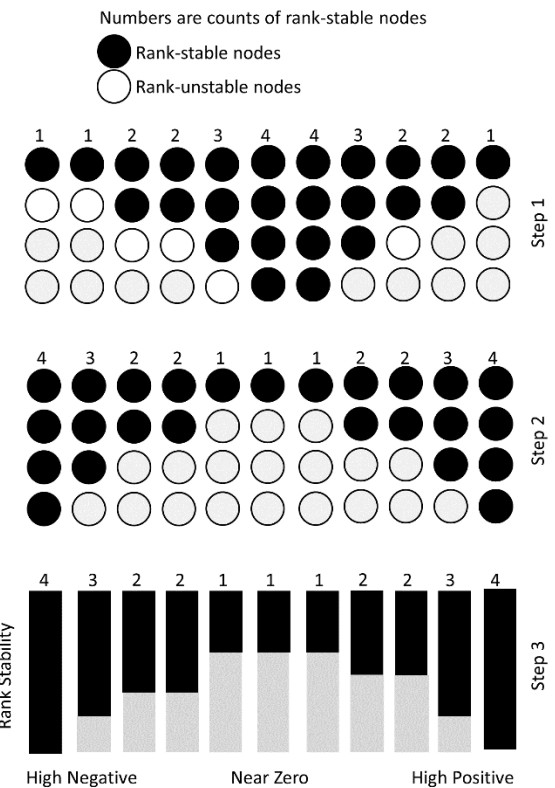

*Figure X. Tree behavior according to node rank stability during a fictitious neural network training progression; number above bars are rank stability values.*