# OpenReview forum: "Explainable Artificial Intelligence: Reaping the Fruits of Decision Trees"
_ICLR.cc/2023/Conference — Submitted to ICLR 2023_

### Official Review · Reviewer_5BfE · 2022-10-23

**Confidence:** 3
**Correctness:** 4
**Technical Novelty And Significance:** 2
**Empirical Novelty And Significance:** Not applicable
**Recommendation:** 5

**Clarity, Quality, Novelty And Reproducibility:**

While the visualization method appears original and new as well as easy to understand and clearly described, it is hard to understand what the practical value of those visualizations is.

**Strength And Weaknesses:**

The paper has a clearly defined experiment setup and meaningful spread of experiments to study a method. I think the paper does not sufficiently discuss how to draw conclusions from the proposed visualizations and summarizations. Adding that would help substantiate the authors claim that this way of looking at neural network training does allow for future improvements.

**Summary Of The Paper:**

The paper studies methods to explain what happens when training neural networks and draw parallels to decision tree behavior through increasing rank stability in the nodes. They study the patterns that emerge during training in a hope to enable future research improving training methods.

**Summary Of The Review:**

The paper proposes a visualization and demonstrates it in various examples, but does not show how to draw practical conclusions from those visualizations to motivate using those visualizations while designing new applications for neural networks.

---

### Official Review · Reviewer_cBc4 · 2022-10-26

**Confidence:** 5
**Correctness:** 2
**Technical Novelty And Significance:** 1
**Empirical Novelty And Significance:** 2
**Recommendation:** 1

**Clarity, Quality, Novelty And Reproducibility:**

Clarity:
The paper is not clear in terms of objective - need some passes to understand

Quality:
The quality of paper needs a serious overhaul

Reproducibility:
Experiments are easy and can can be reproduced

Novelty:
Not clearly listed or evident from reading

**Details Of Ethics Concerns:**

Docoloc plagiarism check is 8% which is compliant.

**Strength And Weaknesses:**

Strengths:
1. The paper gives very detailed information about dataset and the results, and does some prior art coverage

Weakness:
1. The contributions of the paper are not clearly enlisted or self-revealing
2. The gaps wrt prior art in Related work section should be highlighted instead of mere referring them
3. Reduction of dataset size to 100 images and then experiments is going south of experimental result validity
4. converge to at least 0.90 - why this threshold? any ablation?
5. Not clear about the data distribution and balance affecting results
6. how will it behave under adversarial attacks?

**Summary Of The Paper:**

The work does a study with some experiments on standard dataset with the claim indicating that neural networks behaved like a decision tree, wrt rank stability's increase conveying increase in weight's absolute values.


**Summary Of The Review:**

Not suitable for ICLR - need more dedicated work with crisper problem and approach

Suggestions:
1. Abstract should be to the point
2. The related work section should follow the current trend - topic-wise listing of prior art
3. Dataset section can be reduced and key contributions be elaborated more
4. Clarity needed - what specific problem you are trying to solve and give the logic behind the approach


Miscellaneous:
1. Break long sentences in smaller forms
2. mimicking an empty, black background -> what about white (1)
3. should have used latex lines no to review
4. Analyses in this work were based,  according to weight magnitude, etc. - many grammar / sentence construction mistakes
5. quasi-Poisson link function - link?
6. The Tukey’s Honest Significance post-hoc GLM - link?
7. ANOVA - is correct abbrev in place of AOV

---

### Official Review · Reviewer_b5kq · 2022-10-30

**Confidence:** 3
**Correctness:** 2
**Technical Novelty And Significance:** 2
**Empirical Novelty And Significance:** 2
**Recommendation:** 3

**Clarity, Quality, Novelty And Reproducibility:**

The paper is very difficult to read, due a combination of poor writing / non-standard use of terms, a title mostly unrelated to the work, and a poor organization of the narrative (for example, explaining the core of the paper -decision tree behavior of neural networks- only at the end). It is also not entirely clear what are the novel contributions of the paper, as the actual work presented is quite different than what is hinted from the abstract / introduction / related work.

The work should be mostly reproducible, but core details are missing for an exact replica (for example, the authors list the size of the kernels in the convolutional layers, but they do not mention the number of filters, stride, padding, or size of max pooling.

**Strength And Weaknesses:**


Strengths:
- The connection between decision-tree dynamics and neural networks is intriguing, and (if verified and) if it could be exploited, it could be potentially useful.


Major issues:
- The actual work presented in the paper is quite different from the title, the introduction, related work, and most of the abstract. While the narrative on XAI is interesting, the only connection with the work is perhaps in future work.
This may perhaps be due to a misuse of the term XAI / Explainable Artificial Intelligence, but even so, the paper does not provide any way to "reap the fruits of decision trees" either.
- In particular, what do the results tell us about how a neural network has made its decisions?

- The stability criterion set to 90% accuracy seems fairly arbitrary, and it seems it should rather be task specific (what if a network cannot achieve 90% accuracy on a task?). In general, the notion of "stable models" in the paper is confusing; I think the authors tried to determine when the network stopped learning (so that weights ranks would be finalized), but in that case it would have been better to calculate the magnitude of changes to the total weights vector directly, and/or use learning rate annealing to guarantee convergence (and thus rank stability).

- The authors stress the importance of analyzing individual units in a neural network (in the abstract and introduction), yet in the actual work performed they ignore all units except those of the last hidden layer of the network.

- I am not sure how weights are ranked when multiple output units are used instead of a single one: are all connections from all units in the last hidden layer and ALL the output units considered? Or is the analysis performed separately for each output unit?

- The paper is mostly related to the purported 'decision tree dynamics of neural networks', yet the authors never really explain what this means. It is perhaps a concept known in literature that I am not aware of (if so, it should be cited in the paper), but it would be useful to explain it clearly. The concept is developed a bit further at the end of the paper, but still not clearly.
- I also don't understand how node rank stability would be related to the decision tree behavior of the network? Is it because some of the nodes that get 'fixed' early are made to correspond to early decision nodes in a decision tree, etc?

- I am not entirely convinced that the results demonstrate 'decision tree behavior' in neural networks; it seems to me that the observed dynamics are more likely due to gradient descent in high dimensional spaces.
The authors write that "Such nodes [groups] are analogous to the concept of decision tree roots. The timing when those weights and their ranks were established might have varied among individual runs but did show a clear pattern according to groups."
So, the units in those groups may learn earlier or later than others, but at runtime they are all run at the same time, non-hierchically. I understand it may be used as a type of combinatorial code perhaps, but that is not really assessed (e.g., can we actually see what the neurons in the earlier vs later groups encode?). I would be grateful again if the authors could explain the concept more clearly.
- In any case, *I suspect you would see the same exact behavior even in simple logistic regression* (for example, inputs -in this case corresponding to the last hidden layer- more correlated to the classification task would increase in weight earlier, leading to a larger MRS;  isn't this what happens in the paper?  Wouldn't it be easier to study it in this or similar simpler case first?). This experiment would be useful even if the dynamics observed were found to be different, as it could be used to make a stronger argument.


Minor issues:
- Clarity of the text can be improved greatly (mostly organization of the text, but also some language issues and misuse of technical terms);  for example "especially when realizing that weight values may not be as random as previously thought", or "Node analysis consisted of an assessment of whether the progression of nodes from the initial run to stabilization could be modeled by a decision tree, potentially identifying the most significant nodes in determining model accuracy early in the training process.".
- The distinction between 'Convolutional' vs 'Deep Neural Networks' is not entirely correct. While one may imagine a very shallow CNN, they are in practice always deep. In particular, the CNN used in this paper is deeper than the 'DNN' network (3 layers + the same MLP as the other network). Further, what the authors call a DNN is actually a fully-connected MLP.

- Evaluation is performed only on 3 very simple datasets, and only 2 network architectures. It would have been more useful to also include realistic architectures (for example, ResNets), and some simple and well understandable ones (e.g., a simple logistic regression / single-layer nn, as I discussed above).

- In the MLP (called 'DNN' in the paper), the authors mention that each layer has "30x30 nodes"; however, it does not make sense to report the layer sizes like this, as the layers are fully connected (i.e., all spatial arrangement is lost, so whether they are 900 units arranged on a line, a square, or a thorus, it makes no difference).

- "All experimental runs converged after 150 epochs" -> I think you mean "converged within 150 epochs"?


**Summary Of The Paper:**

The authors present an empirical investigation to look for patterns in the connections of the output layer of neural networks that may hint at a behavior akin to decision trees.
The evaluation is performed on 3 simple datasets and 2 network architectures (an MLP + an MLP preceded by 3 convolution+pooling layers). The evaluation consisted in training the neural networks on each task while recording the 'rank stability' of neurons in the last hidden layer, measured as the number of epochs in which each unit maintained its weight rank consecutively until convergence.


**Summary Of The Review:**

I think that the topic of the paper is interesting overall, but it cannot be published in its current form.
The presentation of the paper is confusing, and the work done seems mostly unrelated to the title and introduction. In particular, the actual work presented in the paper has almost nothing to do with explainable AI.

Finally, I am not at all convinced that the results observed lead to the strong conclusions the authors make.

---

### Official Review · Reviewer_ZHzr · 2022-11-01

**Confidence:** 3
**Correctness:** 2
**Technical Novelty And Significance:** 1
**Empirical Novelty And Significance:** 3
**Recommendation:** 3

**Clarity, Quality, Novelty And Reproducibility:**

- The paper is well written, and the goal well explained.
- Several claims lack support (both theoretical and empirical support from the paper's experiments).
- Studying the progression of "node rank" (as defined by the authors) seems interesting and somewhat novel, although I am not familiar with the full literature on this topic.

**Strength And Weaknesses:**

### Strengths
- The paper tackles an under-explored field of XAI research: as the authors point out, it is true that most explainability methods have focused on manipulating input data (e.g., perturbations), and there is a lack of research on the influence of individual (or groups of) neurons.
- The paper is well written, and visualizations are interesting.


### Weaknesses

1. The paper defines "node stability" as
> "the number of epochs wherein nodes held their rank in terms of weight value compared to their rank on the last epoch".

By this definition, stability will always reach 100% in the last epoch, and little can be determined from the "exponential-looking" curve of DNNs stability. The high increase in the end is, by definition, inevitable, and does not really indicate the network's weights are "more stable".
However, the underlying idea of measuring NN stability is interesting. Perhaps it would be better measured as stability of rank between each two consecutive epochs, and not using the last epoch as an anchor.

2. Equivalence between node significance and its weight value is unfounded (although proving this would be interesting).

Throughout the paper the absolute weight value of a node is used equivalently to its "significance" or "influence in performance". This is actually not backed up by theory or by empirical results. A node with high weight value may consistently receive inputs with very low absolute value (depending on the dataset), and a node with low weight value may receive inputs with high absolute value. The relationship between absolute weight value and the influence of that node on model performance is never tested in the paper.

It would indeed be interesting to have an experiment where each node (or group of nodes) is pruned at a time, and performance of the pruned model is assessed, so we could draw some empirical conclusions on whether intermediate nodes are indeed less valuable than higher rank nodes.

3. The use of the term "decision tree", both in the title and throughout the paper, is a bit misleading. No DTs are actually used anywhere in the paper, it is continuously used as (somewhat arbitrary) analogy. Nodes with high absolute weight value do not behave as root nodes do in DTs (or at least this is not proven in the paper).

4. The reader is left wandering what is a possible outcome of discovering "decision tree behavior" in NNs? How can this be used? A lot of hedging is used throughout the paper with respect to possible utility of its results ("may", "might", "can", "possibly", etc.)

**Summary Of The Paper:**

This paper investigates whether neural networks (either CNNs or DNNs) follow "decision-tree-like" behavior. This "decision tree" behavior is defined by the authors as having nodes with high absolute weight value (used as analogous to node significance) determined early in the training process - analogous to root nodes in decision trees.

The paper conducts a series of empirical experiments with 3 datasets and 2 model types (CNN and DNN) to determine if a trend does exist.
The focus of the study is further reduced to the second to last layer (the layer before the softmax output), dubbed "focus layer".

CNNs are found to have higher focus layer "node stability" (defined by the authors as a node maintaining its rank when compared to its rank on the latest train epoch) than DNNs.

**Summary Of The Review:**

The paper does find an interesting quirk of training CNNs vs DNNs: the former's "node stability" progresses more steadily than the latter's. However, it is not clear (a) why this happens, and (b) why does this matter? The use of node rank as a heuristic for pruning those nodes lacks justification. Also, the paper's definition of "node stability" can have a high influence on its conclusions, as it only measures stability between an epoch and the latest train epoch, instead of between each two consecutive epochs.

---

### Decision · Program_Chairs · 2023-01-20

**Decision:**

Reject

**Justification For Why Not Higher Score:**

The reviewers raised reasonable objections (around clarity, around impact, around logical issues), none of which were rebutted by the authors.

**Justification For Why Not Lower Score:**

N/A

**Metareview: Summary, Strengths And Weaknesses:**

This work received four reviews, all negative. The author did not provide a rebuttal to the various issues raised by the reviewers. I suggest the authors revise the paper along the lines of the reviews.

**Summary Of Ac-Reviewer Meeting:**

N/A